# Conversion of mammalian cell culture media waste to microbial fermentation feed efficiently supports production of recombinant protein by *Escherichia coli*

Ciara D. Lynch[ID][1,2], David J. O'Connell[ID][1,2]*

1 BiOrbic, Bioeconomy SFI Research Centre, O'Brien Centre for Science, University College Dublin, Belfield, Dublin, Ireland, 2 School of Biomolecular & Biomedical Science, Conway Institute, University College Dublin, Belfield, Dublin, Ireland

* david.oconnell@ucd.ie

**Data Availability Statement:** The mass spectrometry proteomics data have been deposited to the ProteomeXchange Consortium via the PRIDE [1] partner repository with the dataset

## Abstract

Deriving new value from waste streams through secondary processes is a central aim of the circular bioeconomy. In this study we investigate whether chemically defined spent media (CDSM) waste from cell culture bioprocess can be recycled and used as a feed in secondary microbial fermentation to produce new recombinant protein products. Our results show that CDSM supplemented with 2% glycerol supported a specific growth rate of *E. coli* cultures equivalent to that achieved using a nutritionally rich microbiological media (LB). The titre of recombinant protein produced following induction in a 4-hour expression screen was approximately equivalent in the CDSM fed cultures to that of baseline, and this was maintained in a 16-hr preparative fermentation. To understand the protein production achieved in CDSM fed culture we performed a quantitative analysis of proteome changes in the *E. coli* using mass spectrometry. This analysis revealed significant upregulation of protein synthesis machinery enzymes and significant downregulation of carbohydrate metabolism enzymes. We conclude that spent cell culture media, which represents 100s of millions of litres of waste generated by the bioprocessing industry annually, may be valorized as a feed resource for the production of recombinant proteins in secondary microbial fermentations. Data is available via ProteomeXchange with identifier PXD026884.

## Introduction

Diversion of waste streams generated by bio-industries to secondary processes to produce valuable products through microbial and chemical engineering has become a central pillar of the circular bioeconomy [1]. One example of a waste stream from bioindustry that has yet to be diverted to the creation of new valuable products is the cell culture media used in the bioprocessing of protein drug molecules. The commercial production of monoclonal antibodies using Chinese hamster ovary (CHO) cell bioprocess in 2019 alone resulted in approximately 30 metric tonnes of protein product [2]. The chemically defined cell culture media used to

identifier PXD026884. Submission details: Project
Name: E. coli BL21 Gold LC-MSMS in varying
growth conditions Project accession: PXD026884.

**Funding:** CL & DO'C are funded by the centre for
doctoral training, Atoms-2-Products. The A2P CDT
is supported by the Science Foundation Ireland
(SFI) and the Engineering and Physical Sciences
Research Council (EPSRC) under Grant No. 18/
EPSRC-CDT/3582. The work was also supported
by the Science Foundation Ireland funded BiOrbic
bioeconomy research centre under grant no. 16/
RC/3889. The funders had no role in study design,
data collection and analysis, decision to publish, or
preparation of the manuscript.

**Competing interests:** The authors have declared
that no competing interests exist.

feed the mammalian cells is itself a sophisticated chemical formulation that also represents a
significant waste product in downstream processing. The formulation of chemically defined
media used to culture stable cell lines in bioprocesses has been designed to remove the need
for serum addition to achieve optimal cell growth and facilitate the purification of the
expressed protein [3]. The regulations surrounding the bioprocessing of therapeutic proteins
for drug use requires defined media without the addition of animal ingredients that cannot be
fully standardised. Additionally, stable cell line clones for the expression of commercial pro-
teins relies on genomic integration of the target protein producing genes rather than transient
expression from plasmid constructs. This removes the requirement for selective pressure with
added antibiotics to maintain plasmid constructs [3].

Based on a product titre of 10 g/L of IgG in batch and fed batch systems, with the total yield
of protein product produced in 2019 as a reference, we estimate that up to 300 million litres of
cell culture waste is generated annually [2, 4]. Systematic approaches to valorize this waste as
part of a commitment to a circular bioeconomy have not been investigated. We hypothesise
that spent media from CHO cell culture has the potential to support *E. coli* fermentation and
the production of recombinant protein titres when recycled from bioprocessing systems.

The re-use of spent media has been investigated in certain process systems, mostly through
feeding spent media from the original culture as a supplement to fresh media in the same sys-
tem [5–9]. These have reported some improvement in protein expression and growth rates.
For example, IgG expression titre in mouse hybridoma cell culture was increased by as much
as 50% when the cells were grown in culture media with 33% spent media supplementation
[7]. Conversely, feeding the cells with 100% spent media led to a significant decrease in growth
and protein production by the mammalian cells. This decrease was attributed to the build-up
of auto-inhibitory metabolites and lack of nutrients.

Genetic engineering of a wide array of bacteria has been a central pillar of the development
of the circular bioeconomy, enabling new product development from diverse organic waste
streams including food oils, agrochemical waste and used bioplastics [10–15]. We aimed to
test our hypothesis using *Escherichia coli*, a robust and versatile bacteriological expression host
that is widely used commercially to produce recombinant proteins. The production of recom-
binant human insulin in *E. coli* for example, is a major milestone in human drug production
[16, 17]. *E. coli* has been shown to utilize breakdown metabolites such as lactate present in
spent culture medias [18] and can grow successfully in spent media from the fermentation of
other cell types [9].

Comparative analysis of culture growth rates of an E. coli BL21 strain harbouring a recom-
binant mCherry-EF2 expression construct in rich microbiological media (LB) and in chemi-
cally defined spent media (CDSM) that had previously been used to culture CHO cells, either
with or without supplementation, confirmed CDSM as a viable feed for bacterial fermentation.
Analysis of recombinant protein production by the cultures confirmed equivalent recombi-
nant protein titres post-purification when compared to LB.

## Results

### Growth rate analysis of *E. coli* BL21 mCherry/EF2 in minimal media, baseline rich media and chemically defined spent media

Growth rates of *E. coli* cultures grown in M9 minimal media with carbon source supplementa-
tion were measured to determine the optimal supplementation conditions that could then be
applied to the CDSM fed cultures. Glycerol was selected as a suitable carbon supplementation
as it is also a waste product from other processes. Gradients of glycerol additions were tested at
concentrations of 1%-3% (Fig 1A). 2% glycerol supplementation was shown to produce a

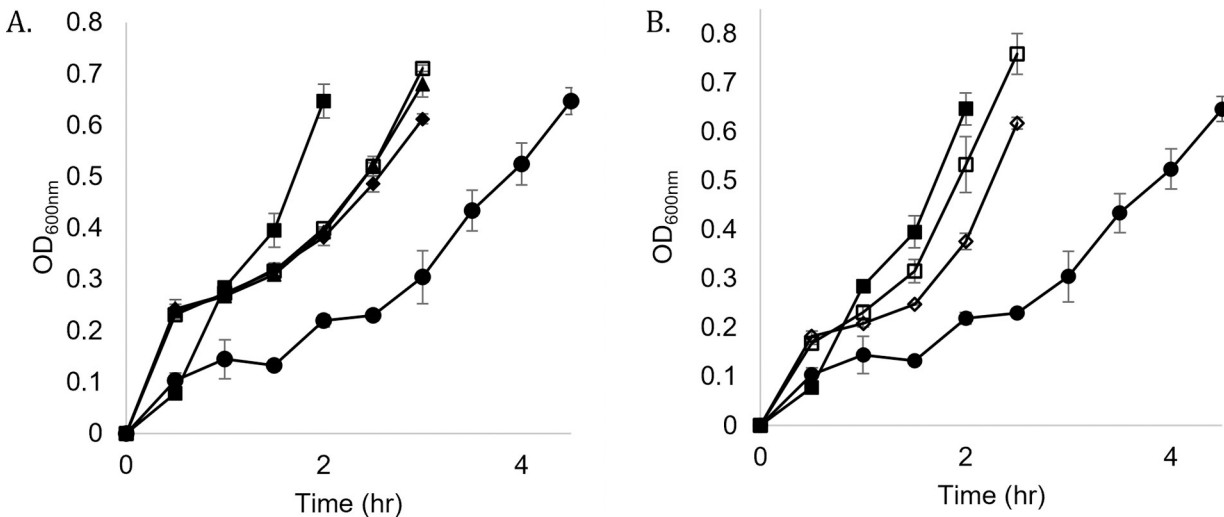

**Fig 1. _E. coli_ culture growth rates in bacteriological and cell culture media. (A).** Glycerol supplementation in M9 minimal media; ■ = baseline media, ● = minimal media, ◆ = minimal media + 1% glycerol, □ = minimal media + 2% glycerol, ▲ = minimal media + 3% glycerol. **(B).** Unsupplemented and supplemented chemically defined spent media (CDSM). ■ = baseline media, ● = M9 media, ◇ = chemically defined spent media (CDSM) unsupplemented, □ = CDSM with 2% glycerol. All time points were completed in triplicate with standard deviation as error bars. Numerical data used to generate growth curves and standard deviations are reported in S2 Table.

specific growth rate of 0.673 generations per hour, as compared to the growth rates of 1% and 3% at 0.578 and 0.647.

The chemically defined spent medium used was CHOgro® spent media harvested from CHO (Chinese Hamster Ovary) cell culture. The specific growth rate of the _E. coli_ in the CDSM was 0.704 gen/hour, approximately 70% of the rate achieved in the baseline LB media. Supplementing CDSM with 2% glycerol led to a further increase in growth rates to ~94% of that of LB (Fig 1B). This supplementation was chosen for all medias tested for protein expression analysis.

## Protein expression analysis in chemically defined spent media fermentation

Recombinant mCherry-EF2 expression was tested in cultures grown in CDSM supplemented with 2% glycerol and the baseline LB with 2% glycerol. A four-hour expression screen post-induction with IPTG showed that protein production in CDSM was equivalent to baseline LB, with an average yield of 100.58 mg/L compared to LB's 92.57 mg/L (Fig 2A).

We next tested a 16-hour expression post-induction with IPTG using CDSM, and included a second chemically defined spent medium, Expi-CHO® (termed CDSM II), to examine the robustness of our finding with widely used commercial medias. Protein yield from the CDSM I fed culture closely matched the yield of that from LB with a protein yield of 159.82 mg/L compared to 168.92 mg/L (Fig 2A). The alternate spent media type tested, CDSM II, managed a successful yield of 127.57 mg/L. It should be noted that the CDSM II condition also lost some yield during expression, seen by the presence of the recombinant protein in the extracellular media also. SDS-PAGE analysis confirmed that the monomeric peak taken from SEC corresponding to the mCherry-EF2 protein at 32 kDa.

## Mass spectrometry analysis of E. coli cultures at the whole proteome level

We performed a proteomic analysis of the protein expression patterns in the bacterial cultures themselves using LC-MS/MS. The analysis of proteomic data files was carried out using

A.

| Measurement | LB + 2% Glycerol | CDSM I + 2% Glycerol | CDSM II + 2% Glycerol |
|---|---|---|---|
| Specific growth rate (k) | 1.07 | 0.94 | 0.92 |
| R-squared coefficient | 0.97 | 0.95 | 0.95 |
| 4-hour biomass (g) (±SD)* | 0.25 (± 0.02) | 0.43 (± 0.04) | **nd** |
| 4-hour protein yield (mg/L) (±SD)* | 92.57 (± 19.98) | 100.58 (± 6.77) | **nd** |
| 16-hour biomass (g) (±SD)* | 0.46 (± 0.02) | 0.47 (± 0.11) | 0.56 (± 0.06) |
| 16-hour protein yield (mg/L) | 168.92 | 159.82 | 127.57 |

B.

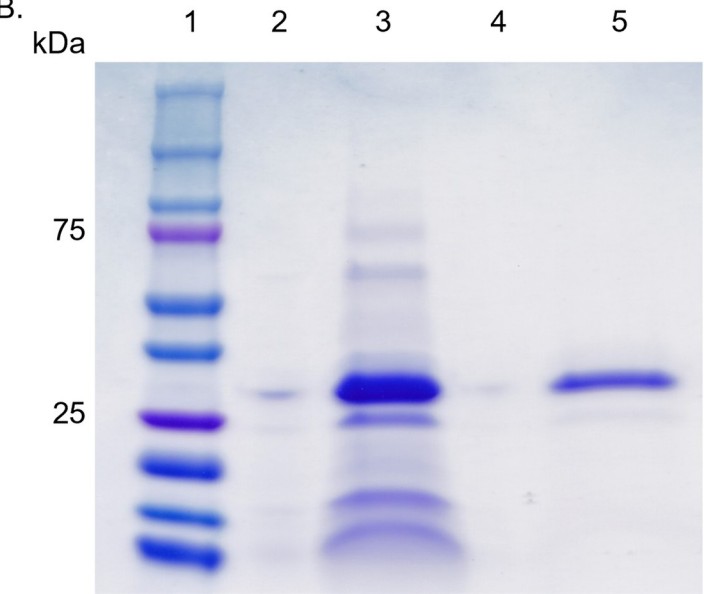

**Fig 2. Protein expression and purification analysis. (A.)** Table of growth characteristics, biomass, and protein yields. LB media; Lysogeny Broth baseline rich media. M9 media; minimal salts media. CDSM I; chemically defined spent media CHOgro®. CDSM II; chemically defined spent media Expi-CHO®. $R^2$ coefficient is the percentage of variability in the growth curve dataset that is accounted for by linear correlation between the OD600 (nm) and the time (hr). For the equation for the specific growth rate, see Methods. * n = 2 for all standard deviation calculations. **(B.)** SDS-PAGE analysis of purified protein versus input supernatant sample. Lane 1 = protein marker, lane 3 = post-boil supernatant sample from CDSM, lane 5 = purified monomeric fraction from size exclusion chromatography of CDSM. Full image available as S1 Raw images.

MaxQuant. Using the protein expression pattern for *E. coli* grown in LB media as a baseline reference, a student's t-test with a false discovery rate of <0.05 identified 655 differentially expressed proteins in *E. coli* cultures grown in unsupplemented CDSM. In 2% glycerol supplemented CDSM conditions there were 167 proteins differentially expressed by comparison (Fig 3).

In CDSM I supplemented with 2% glycerol, 1,222 proteins were found to be dysregulated and of these, 167 proteins were statistically significant. From these 167 proteins, 87 proteins were identified as upregulated and 80 proteins as downregulated. A program that measures the quantitative differences in expression by difference in LFQ intensity was written using Python script and using a cut-off significance value of >0.5 or <-0.5 in intensity identified the

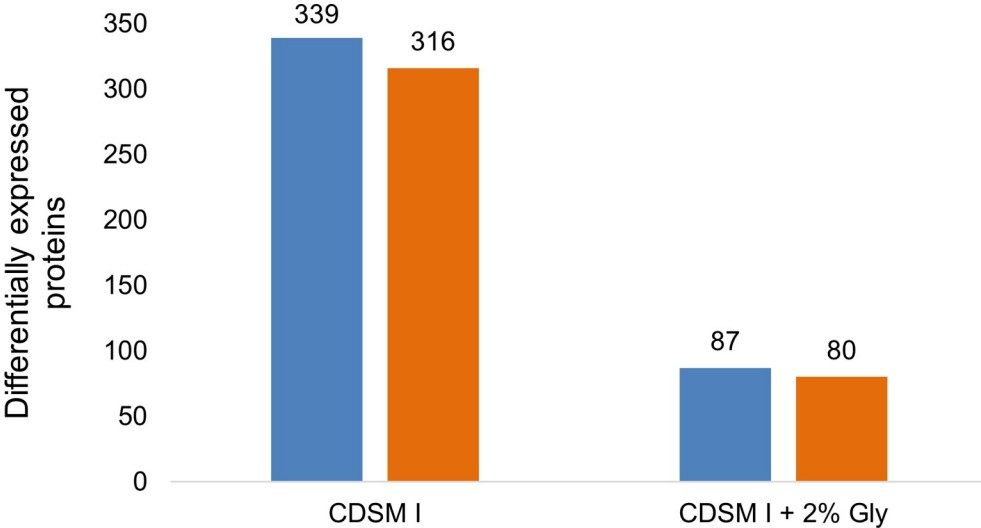

**Fig 3. Number of statistically significant differentially expressed proteins compared to the baseline condition in LB media.** Orange bars represent the number of downregulated proteins while blue represents all upregulated proteins compared to baseline LB media after a student's t-test with an FDR of $< 0.05$.

most significantly changed expression levels, shown here for the first time as a pool table plot (Fig 4). Proteins upregulated in the CDSM I + 2% glycerol condition were principally in the amino-acid and purine biosynthesis pathways, while proteins that were most significantly downregulated were in the carbohydrate metabolism pathway (Fig 4).

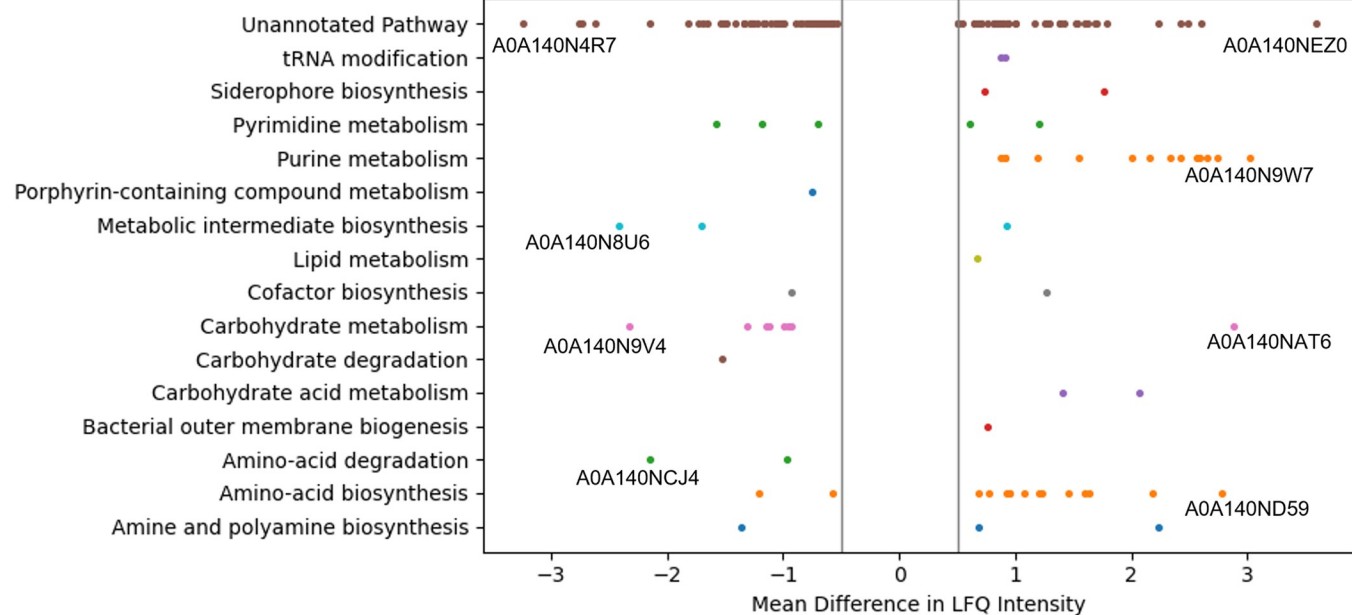

**Fig 4. Pool-table plot of the proteomic analysis of the significantly dysregulated proteins from the sample condition of CHOgro® with 2% glycerol supplementation.** This graph was constructed using the Matplotlib library in Python. Significant upregulation of expression is shown to the right of the central threshold divider (-0.5 to 0.5), with significant downregulation to the left. Significance in this case was defined as having a mean difference in LFQ intensity compared to the baseline media of more than 0.5 in either direction after a student's t-test and with an FDR of <0.05. The top three pathway proteins and the top one unannotated pathway protein are labelled directly below their representative point, for both up- and down-regulated.

Amino acid biosynthesis and purine metabolism functional pathways were up-regulated in all cultures grown in CDSM compared to the baseline. Carbohydrate metabolism by contrast was mainly downregulated. Within amino acid biosynthesis are some of the most statistically significantly upregulated enzymes, such as diaminopimelate decarboxylase and Aspartate-ammonia ligase, two enzymes that are upregulated with a mean difference of LFQ intensity of $> 2$, along with 10 other significantly upregulated enzymes in this pathway. Purine metabolism also features highly upregulated enzymes such as glutamine dependent amidophosphoribosyltransferase, with a difference in intensity of $> 3$ along with 12 other significantly upregulated enzymes.

## Discussion

### Can spent media from CHO cell culture be reused to feed *E. coli* fermentation?

The creation of chemically defined culture media has led to increasing recombinant protein titres and protein quality [19, 20]. These successful developments have resulted in this cell culture media becoming a significant waste stream with approximately 300 million litres sent for disposal annually [2].

Our data indicates that chemically defined spent media (CDSM) is a potentially valuable resource for producing new recombinant proteins when compared with microbiological media prepared with casein digests and yeast extract, such as the LB media studied here. In this study we investigated the expression of a recombinant fusion protein ligand (EF2) derived from mCherry and the calcium binding protein Calbindin D9k. This recombinant protein has been developed as a ligand for a highly specific and high affinity purification system [21–24]. The growth rate of the expression culture in the CDSM + 2% glycerol reached was similar to the LB at ~94% while the 4-hour protein yield was approximately equal, a striking and unanticipated finding (Fig 2A). This finding was further verified in the 16-hour fermentation, a model of a preparative scale expression, whereby the CDSM cultures successfully supported a protein expression over this longer period with approximately the same yield (159.82 mg/L) as those cultures grown in the nutritionally rich LB media (168.92 mg/L) (Fig 2A). This finding prompted us to investigate the proteome of the cultures to identify proteins responsible for metabolic changes that may contribute to this phenotype.

### Proteomic analysis

Out of the 629 upregulated proteins identified in the optimised condition of CDSM + 2% Glycerol, the most statistically significantly upregulated proteins are enzymes involved in the amino acid biosynthesis pathway. For example, both Diaminopimelate decarboxylase and Aspartate-ammonia ligase are two highly upregulated proteins with LFQ differences of $> 2$ (Fig 4, Table 1), involved in lysine biosynthesis and asparagine biosynthesis pathways that are dependent on glutamine uptake [25]. Glutamine (4 mM) is a supplement added to the chemically defined media prior to CHO cell culture. Amino acid accumulation such as increased L-asparagine is a known feature of the *E. coli* stress response, to make available the building blocks needed for synthesis of stress response proteins [26]. This upregulation may indicate the activation of a stress response in the cultures that is not activated in rich microbiological media but that contributes to recombinant protein expression. Other highly upregulated proteins ($>2$ fold) include glycerol metabolism enzymes such as Phosphogluconate dehydratase, purine metabolism enzymes such as glutamine dependent Amidophosphoribosyltransferase, and iron uptake proteins such as Enterobactin non-ribosomal peptide synthetase EntF, among

**Table 1. Most significantly dysregulated *E. coli* proteins and the associated annotated functional pathway.** Proteins highlighted by accession number in Fig 4 are described here with one example of an upregulated and downregulated protein lacking an annotated functional pathway.

| Uniprot Entry | Protein Name | Gene | Pathway | Description | LFQ Difference |
|---|---|---|---|---|---|
| **Up-regulated Proteins** | | | | | |
| A0A140NEZ0 | Periplasmic copper-binding protein | cusF | Unannotated | Homologue of Cation efflux system protein | 3.59 |
| A0A140N9W7 | Amidophosphoribosyltransferase | purF | Purine metabolism | IMP biosynthesis via de novo pathway | 3.02 |
| A0A140NAT6 | Phosphogluconate dehydratase | edd | Carbohydrate metabolism | Entner-Doudoroff pathway; catalyses glucose into pyruvate | 2.89 |
| A0A140ND59 | Aspartate—ammonia ligase | asnA | Amino-acid biosynthesis | L-asparagine biosynthesis | 2.79 |
| **Down-regulated Proteins** | | | | | |
| A0A140N4R7 | Protein YcfR | yhcN | Unannotated | Homologue of Peroxide/acid stress response protein YhcN | -3.24 |
| A0A140N8U6 | Phospho-2-dehydro-3-deoxyheptonate aldolase | aroF | Metabolic intermediate biosynthesis | Member of the chorismate biosynthesis pathway | -2.42 |
| A0A140N9V4 | Probable malate: quinone oxidoreductase | mqo | Carbohydrate metabolism | Tricarboxylic acid cycle; oxaloacetate from (S)-malate | -2.33 |
| A0A140NCJ4 | D-amino acid dehydrogenase | dadA | Amino-acid degradation | D-alanine degradation | -2.14 |

others [27–29]. Interestingly the increased expression of the glycerol metabolism enzyme Phosphogluconate dehydratase and glutamine dependent Amidophosphoribosyltransferase correlates well with the supplementation with both glycerol and glutamine and confirms the sensitivity of this MS analysis.

Among the 595 downregulated proteins in the CDSM + glycerol condition are enzymes involved in the TCA cycle such as the probable malate:quinone oxidoreductase, some specific stress response proteins such as protein YcfR (an acid stress response protein), and nucleotide synthesis/salvage proteins, such as cytidine deaminase. Many of these proteins depend on the presence of glucose in order to be active, leading to a possible reason for their downregulation in the CDSM + glycerol condition [30, 31]. Other downregulated proteins which are statistically significant include D-amino acid dehydrogenase, involved in amino-acid degradation, and generally present in high levels in LB rich broth due to extracellular D-amino acids which are not present in the CDSM [32, 33].

One of the main metabolic pathways that was upregulated in the CDSM media compared to LB was the amino-acid biosynthesis pathway. The baseline condition of LB contains tryptone, a source of nitrogen-containing peptides for *E. coli* growth, whereas the spent CHOgro® media must rely on the proteins released by the CHO cell culture conditions [34]. CHO cell growth in chemically defined media produces metabolites such as ammonium and lactate that can inhibit further growth [35]. These factors however can be used by *E. coli* to grow, utilising ammonium as its preferred nitrogen source and can break down lactate as a possible carbon source if needed [36]. The 102 proteins that are significantly dysregulated and are yet to be assigned a pathway on Uniprot are involved in a wide array of metabolic processes in the *E. coli* cell, such as the acid stress response protein seen to be the most downregulated in Table 1. Further study into these individual proteins will help to understand the proteomic adaptions undertaken by the CDSM fed *E. coli*.

### Proteomic analysis of the nutritional content of CDSM

CHO cells generally produce approximately 1,400 host cell proteins (HCPs) which are detectable in the clarified spent culture media, with ~80% of the top 1000 of these HCPs in common across multiple cell lines [37, 38]. Mass spectrometry-based analysis has been widely employed to characterise HCP's across a number of studies [39–41].

Mass spectrometry analysis of the CDSM alone identified 879 host cell proteins from CHO in the spent media after culturing (see S1 Table). These proteins were identified from all cellular compartments suggesting they are debris from CHO cell lysis in addition to any active secretion by the growing cells. These host cell proteins represent a source of amino acid building blocks for the increased recombinant protein production capabilities of the *E. coli* cultures.

## Conclusion

We have shown in this study that mammalian cell culture waste, the chemically defined synthetic media used to grow Chinese Hamster Ovary cells, is conditioned such that it provides a nutritious feed for the growth of *E. coli* cultures in secondary fermentation. The growth rate of the culture in this waste medium is similar to that of rich microbiological culture media and upon supplementation with another waste by-product, glycerol, the growth rate is enhanced. Importantly, the expression of a recombinant protein from an expression plasmid construct is seen to be equivalent in protein titre between 4 to 16 hours showing that this waste has a real value in a biotechnological context. This approach may be further developed based on a deeper understanding of the protein expression patterns analysed here, that show significant upregulation and downregulation of metabolic enzymes and pathways. This approach can begin a route for the capture of a bioprocessing waste stream for the circular bioeconomy.

## Experimental procedures

### Mammalian cell culture

Chinese Hamster Ovary (CHO) cells were incubated in 20 ml of serum-free CHOgro® Expression media supplemented with 4 mM L-Glutamine in T75 adherent cell line flasks, at 37˚C with 5% $CO_2$. Cells were split at 70–80% confluency every 3–4 days. Spent media was harvested at each split and was clarified of cells and cellular debris by centrifugation at 300 x g for 4 minutes prior to storing at 4˚C for up to 14 days. The pH of the spent media was generally > 7.5 post-culture and this was adjusted to pH 7 prior to use in bacterial fermentation with addition of dilute HCl. No other supplementation was added. Remaining glucose levels in the spent media after harvesting was 1.6%.

### Growth and expression cultures of *E. coli*

*E. coli* BL21 Gold was transformed with an mCherry-EF2 fusion protein expression construct with a T7 promoter expression system. A 10 ml starter culture of BL21 transformed *E. coli* was grown by incubating a single, isolated colony in Lysogeny Broth (LB) media, 2% glucose, and 100 µg/ml ampicillin. The culture was grown in a shaking incubator at 250 rpm, 37˚C for 16 hours. Growth curves were taken from 50 ml cultures grown in triplicate in 250ml flasks by measuring $OD_{600}$ every half an hour for growth curve plotting until the $OD_{600}$ reached 0.6. The four-hour expression was carried out in duplicate 50 ml cultures with 2% glycerol and a 1 in 20 dilution of starter culture in the expression media in a shaking incubator at 250 rpm, 37˚C.Expression of the fusion protein was induced by addition of 1 mM IPTG after $OD_{600}$ reached 0.6. Cultures were then incubated on a shaking incubator at 250 rpm, 30˚C for 4 hours or for 16 hours for the overnight expression experiment. Cultures were then spun at

4˚C, 4000 x g for 20 minutes to harvest the pellets which were weighed prior to protein purification.

## Calculations of specific growth rate

$$k = \frac{log_{10}[X_t] - log_{10}[X_0]}{0.301 \times t}$$

Specific growth rate (k) was calculated by the above equation, where k is generations per hour and t is time in hours. Plots of growth curves of $OD_{600}$ (nm) versus time (hrs) were generated to gather this data in Excel. $X_t$ is one $OD_{600}$ value at a later position, and $X_0$ is another $OD_{600}$ value taken from an earlier position using a trendline.

## Recombinant protein purification

Cell pellets were re-suspended using a buffer containing 10 mM Tris and 2 mM $CaCl_2$ pH 7.4 before lysing by sonication. The lysate was then boiled at 85 degrees Celsius and spun at 15,000 x gfor 30 minutes. Supernatant was harvested and referred to as the post-boil (PB) sample. 0.5 ml of this PB was run on a Superdex 200 10/300 size exclusion chromatography column, using Hepes Buffer Saline as a running buffer. Protein concentration was measured with a DeNovix DS-11 Spectrophotometer, using the UV-Vis application. Measurements were taken at 585 nm for mCherry yield, with an extinction coefficient calculated at 44,854.2 $M^{-1}cm^{-1}$ and a protein molecular weight of 30,667.4 g/mol.

## Proteomic analysis

4 ml of *E. coli* culture was harvested after reaching $OD_{600}$ of 0.6, but before IPTG induction. These cultures were re-suspended in 8M urea. 5mM DTT was then added, and samples were incubated at room temperature for 10 minutes. After incubation, 10mM iodoacetamide was added, and samples were incubated in the dark for 10 minutes at room temperature. Samples were brought to neutral pH by addition of 150 μl of 200 mM $NH_4HCO_3$. Samples were then digested with trypsin at 37˚C and 300 rpm on a thermomixer overnight. Samples were spun at 4000 x g for 2 mins to prevent ZipTip® from getting blocked in the next steps. Digested peptides were filtered through ZipTip® columns and eluted using 70% acetonitrile in acidic water (0.1% formic acid). The elution was dried off by speed-vac at 45˚C. The dried peptides were then re-suspended in 30 μl Buffer A (5% acetonitrile and 0.1% TFA). Analysis of proteomic data files was carried out using MaxQuant [42] to assemble peptides using the *E. coli* BL21 proteome from Uniprot (UP000002032). The software Perseus [43] was used to analyse the resulting dataset for student's t-tests and heatmap generation. A Python programme was generated to give the remaining graphs, using the t-test results generated from Perseus and annotation information from Uniprot for *E. coli* strain BL21 (available on GitHub at https://github.com/Ciara-Lynch/Mass_Spec_Analysis.git).

## Supporting information

**S1 Raw images. Raw gel image for SDS-PAGE.** PDF of uncropped gel image from Fig 2, panel B.
(PDF)

**S1 Table. Table of CHO host cell proteins detected by LC-MS/MS in a spent CHOgro® media sample.** Completed in triplicate with mean LFQ values displayed in third column.
(XLSX)

**S2 Table. Table of numerical data used to generate growth curves and standard deviations.** Completed in triplicate. Corresponds to growth curves generated for Fig 1.
(XLSX)

## Acknowledgments

Thank you to Eugene Dillon and Siobhan Kelly for guidance on mass spectrometry usage.

## Author Contributions

**Conceptualization:** David J. O'Connell.

**Data curation:** David J. O'Connell.

**Formal analysis:** David J. O'Connell.

**Funding acquisition:** David J. O'Connell.

**Investigation:** Ciara D. Lynch, David J. O'Connell.

**Methodology:** Ciara D. Lynch, David J. O'Connell.

**Project administration:** David J. O'Connell.

**Resources:** David J. O'Connell.

**Supervision:** David J. O'Connell.

**Validation:** David J. O'Connell.

**Writing – original draft:** Ciara D. Lynch, David J. O'Connell.

**Writing – review & editing:** Ciara D. Lynch, David J. O'Connell.

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
