## [Decision Letter · Decision Letter 0]

4 Mar 2022

PONE-D-21-40475Conversion of mammalian cell culture media waste to microbial fermentation feed efficiently supports production of recombinant protein by Escherichia coliPLOS ONE

Dear Dr, O'Connell

Thank you for submitting your manuscript to PLOS ONE. After careful consideration, we feel that it has merit but does not fully meet PLOS ONE’s publication criteria as it currently stands. Therefore, we invite you to submit a revised version of the manuscript that addresses all the points raised during review process. It is important to comment on the choice of culture media utilized and the impact of the antibiotics on microbial physiology. Is the efficacy of the antibiotics diminished in the spent fluids? Please submit your revised manuscript by Apr 18 2022 11:59PM. If you will need more time than this to complete your revisions, please reply to this message or contact the journal office at plosone@plos.org. Please include the following items when submitting your revised manuscript:A rebuttal letter that responds to each point raised by the academic editor and reviewer(s). You should upload this letter as a separate file labeled 'Response to Reviewers'.A marked-up copy of your manuscript that highlights changes made to the original version. You should upload this as a separate file labeled 'Revised Manuscript with Track Changes'.An unmarked version of your revised paper without tracked changes. You should upload this as a separate file labeled 'Manuscript'.

We look forward to receiving your revised manuscript.

Kind regards,

Vasu D. Appanna

Academic Editor

PLOS ONE

Journal Requirements:

“The authors wish to thank Science Foundation Ireland and the EPSRC for joint funding of Ms Ciara Lynch through the Centre for Doctoral Training – Atoms to Products. The A2P CDT is supported by the Science Foundation Ireland (SFI) and the Engineering and Physical Sciences Research Council (EPSRC) under Grant No. 18/EPSRC-CDT/3582. The work was also supported by the Science Foundation Ireland funded BiOrbic bioeconomy research centre under grant no. 16/RC/3889.”

“CL & DO'C are funded by Science Foundation Ireland (SFI) and the Engineering and Physical Sciences Research Council (EPSRC) under Grant No. 18/EPSRC-CDT/3582.

CL & DO'C are funded by Science Foundation Ireland funded BiOrbic bioeconomy research centre under grant no. 16/RC/3889.”

“CL & DO'C are funded by Science Foundation Ireland (SFI) and the Engineering and Physical Sciences Research Council (EPSRC) under Grant No. 18/EPSRC-CDT/3582.

CL & DO'C are funded by Science Foundation Ireland funded BiOrbic bioeconomy research centre under grant no. 16/RC/3889.”         

7. In your Data Availability statement, you have not specified where the minimal data set underlying the results described in your manuscript can be found. PLOS defines a study's minimal data set as the underlying data used to reach the conclusions drawn in the manuscript and any additional data required to replicate the reported study findings in their entirety. All PLOS journals require that the minimal data set be made fully available. For more information about our data policy, please see http://journals.plos.org/plosone/s/data-availability.

Reviewers' comments:

Reviewer's Responses to Questions

**Comments to the Author**

1. Is the manuscript technically sound, and do the data support the conclusions?

Reviewer #1: Yes

Reviewer #2: Yes

2. Has the statistical analysis been performed appropriately and rigorously? 

Reviewer #1: No

Reviewer #2: Yes

3. Have the authors made all data underlying the findings in their manuscript fully available?

Reviewer #1: Yes

Reviewer #2: Yes

4. Is the manuscript presented in an intelligible fashion and written in standard English?

Reviewer #1: Yes

Reviewer #2: Yes

5. Review Comments to the Author

Reviewer #1: This is a very interesting study demonstrating the conversion of mammalian cell culture waste to microbial fermentation feed, promoting bacterial growth and recombinant protein production. The authors further found that spent media would upregulate protein synthesis machinery enzymes but downregulate carbohydrate metabolism enzymes. The authors proposed that this approach can begin a route for the capture of a bioprocessing waste stream for the circular bioeconomy. Although this is an interesting topic for the readership of the journal, a significant revision is needed. Below are more detailed comments that need to be addressed.

The culture of mammalian cells are often maintained at 37˚C with 5% CO2 and atmospheric concentration of O2 (21%). It is not clear why this study used 95% O2 instead.

One more, normal culture of mammalian cells needs 10-20% serum, however in this study, serum-free medium was used throughout the whole process.

Some basic information on the spent media needs to be provided, such as pH, with phenol red or not, glucose level, ..

Statistics analysis was missing in all figures.

In the supplemental materials, only upregulated protein synthesis machinery enzymes were listed. It is suggested to list the downregulated carbohydrate metabolism enzymes as well.

Reviewer #2: The idea of using the waste of a cell culture medium as a medium for the growth of bacteria such as the case of E. coli to produce proteins of interest is original and scientifically interesting. The manuscript is very well written. Just a few comments that need to be clarified by the authors before publication.

- The use of antibiotics is often indicated in cell culture media. Residues of these antibiotics present in waste cell culture media could inhibit bacterial growth. In the experimental section of this study, the authors used ampicillin at 100 µg/mL in the culture medium of E. coli but not in the culture medium of the CHO cell line. Could you clarify.

- Do you have an idea of the degree of purity of target proteins in the two types of culture media?

6. PLOS authors have the option to publish the peer review history of their article (what does this mean?). If published, this will include your full peer review and any attached files.

Reviewer #1: No

Reviewer #2: No

---

## [Author Response · Author response to Decision Letter 0]

14 Mar 2022

Dear Dr Appanna,

We thank you and the reviwers for the positive review of our manuscript “Conversion of mammalian cell culture media waste to microbial fermentation feed efficiently supports production of recombinant protein by Escherichia coli”, tracking number PONE-D-21-40475. We have addressed the review comments and have provided a point by point response below.

Editor’s comments:

1. Please ensure that your manuscript meets PLOS ONE's style requirements, including those for file naming. We have made style changes as requested, including changes to reference formatting and author affiliations, addition of continuous line numbering, and removal of figures for inclusion as separate file elements.

2. Please note that funding information should not appear in the Acknowledgments section or other areas of your manuscript. We will only publish funding information present in the Funding Statement section of the online submission form. The funding information has been deleted from the Acknowledgements section of the manuscript. We wish to amend our Funding Statement to the following: “CL & DO'C are funded by the centre for doctoral training, Atoms-2-Products. The A2P CDT is supported by the Science Foundation Ireland (SFI) and the Engineering and Physical Sciences Research Council (EPSRC) under Grant No. 18/EPSRC-CDT/3582. The work was also supported by the Science Foundation Ireland funded BiOrbic bioeconomy research centre under grant no. 16/RC/3889.”

3. We note that you have included the phrase “data not shown” in your manuscript. Unfortunately, this does not meet our data sharing requirements. The sentence in question refers to data that is not a core part of the research being presented and the phrase has been removed from the manuscript.

4. Please state what role the funders took in the study. The funders had no role in this study so please include the amended statement "The funders had no role in study design, data collection and analysis, decision to publish, or preparation of the manuscript."

5. PLOS ONE now requires that authors provide the original uncropped and unadjusted images underlying all blot or gel results reported in a submission’s figures or Supporting Information files. The uncropped original gel result file has been added to our submission as the file S1_raw_images.pdf, and listed in the Supporting Information. The statement ‘Full image available as S1 file in Supporting Information.’ has been added to the text in the legend for figure 2.

6. We note that you have stated that you will provide repository information for your data at acceptance. Should your manuscript be accepted for publication, we will hold it until you provide the relevant accession numbers or DOIs necessary to access your data. If you wish to make changes to your Data Availability statement, please describe these changes in your cover letter and we will update your Data Availability statement to reflect the information you provide. We would like to add the following statement to our Data Availability Statement: “The mass spectrometry proteomics data have been deposited to the ProteomeXchange Consortium via the PRIDE [1] partner repository with the dataset identifier PXD026884.” We have also added this statement to our abstract: "Data is available via ProteomeXchange with identifier PXD026884." Submission details:

Project Name: E. coli BL21 Gold LC-MSMS in varying growth conditions

Project accession: PXD026884

Project DOI: Not applicable

Reviewer account details:

Username: reviewer_pxd026884@ebi.ac.uk

Password: RoWSbofL

7. In your Data Availability statement, you have not specified where the minimal data set underlying the results described in your manuscript can be found. Minimal data set has been added for the growth curve generation as supporting information, under file name S3_table.xlsx. Other minimal data sets are already present, under S2_table.xlsx and on the ProteomeXchange for the raw proteomic data.

Reviewers’ comments:

Reviewer 1:

1. The culture of mammalian cells are often maintained at 37˚C with 5% CO2 and atmospheric concentration of O2 (21%). It is not clear why this study used 95% O2 instead. Thank you for identifying this error in our Experimental Procedures section, which did not correctly indicate that atmospheric concentration of O2 (21%) was used. We have adjusted the text to now read ‘Chinese Hamster Ovary (CHO) cells were incubated in 20 ml of serum-free CHOgro® Expression media supplemented with 4 mM L-Glutamine in T75 adherent cell line flasks, at 37˚C with 5% CO2.’.

2. One more, normal culture of mammalian cells needs 10-20% serum, however in this study, serum-free medium was used throughout the whole process. A sentence and new citation have been added to our introduction explaining the usage of serum-free medium in bioprocess industry along with appropriate citation as follows ‘The formulation of chemically defined media used to culture stable cell lines in bioprocesses has been designed to remove the need for serum addition to achieve optimal cell growth and facilitate the purification of the expressed protein [3]. The regulations surrounding the bioprocessing of therapeutic proteins for drug use requires defined media without the addition of animal ingredients that cannot be fully standardised.’

3. Some basic information on the spent media needs to be provided, such as pH, with phenol red or not, glucose level. A sentence has been added to the Experimental Procedures stating ‘The pH of the spent media was generally > 7.5 post-culture and this was adjusted to pH 7 prior to use in bacterial fermentation with addition of dilute HCl. No other supplementation was added. Remaining glucose levels in the spent media after harvesting was 1.6%.’.

4. Statistics analysis was missing in all figures. Figure 1 contains standard deviation error bars from triplicate measurements stated in the figure legend as follows ‘All time points were completed in triplicate with standard deviation as error bars.’ We have also updated the legend with the following statement ‘Numerical data used to generate growth curves and standard deviations are reported in S3 table in Supporting Information.’. In panel A of Figure two we report standard deviations for duplicate measurements and is mentioned in the legend as follows ‘n = 2 for all standard deviation calculations.’. Figures 3 and 4 report data after a student’s t-test with a false discovery rate of <0.05 as stated in the figure legends. All of these statistical methods are described in the Experimental Procedures. 

5. In the supplemental materials, only upregulated protein synthesis machinery enzymes were listed. It is suggested to list the downregulated carbohydrate metabolism enzymes as well. The supplemental material S2_table is a table of host cell proteins from CHO cell culture detected by mass spectrometry analysis of spent media alone prior to use as a bacterial fermentation feed. This provides detail on the protein content of the spent media as a feed. We have updated the text in the discussion section as follows ‘Mass spectrometry analysis of the CDSM alone identified 879 host cell proteins from CHO in the spent media after culturing (see Table S2).’.The proteomic analysis of E.coli dysregulated proteins including the protein synthesis machinery enzymes and the carbohydrate metabolism enzymes are included as raw data on the ProteomeXchange database with identifier PXD026884. The most significantly upregulated and downregulated proteins involved in protein synthesis and carbohydrate metabolism are shown in table 1 in the main manuscript.

Reviewer 2:

1. The use of antibiotics is often indicated in cell culture media. Residues of these antibiotics present in waste cell culture media could inhibit bacterial growth. In the experimental section of this study, the authors used ampicillin at 100 µg/mL in the culture medium of E. coli but not in the culture medium of the CHO cell line. Could you clarify. We have added a paragraph into our introduction to clarify this as follows ‘Additionally, stable cell line clones for the expression of commercial proteins relies on genomic integration of the target protein producing genes rather than transient expression from plasmid constructs. This removes the requirement for selective pressure from added antibiotics to maintain plasmid constructs.’. 

2. Do you have an idea of the degree of purity of target proteins in the two types of culture media? In figure 2 panel B, the SDS gel with the pre-purified and post-purified protein fractions loaded on lanes 3 and 5 respectively shows a purification of approximately 95% post-size exclusion chromatography which was the typical result from either media type.

We have also added a new reference to our bibliography cited as reference [3] and have updated the order of the citations accordingly.

We hope that with these responses our manuscript may now be published in PLOS ONE. Thank you very much for this review.

Ciara Lynch and David O’Connell

---

## [Decision Letter · Decision Letter 1]

30 Mar 2022

Conversion of mammalian cell culture media waste to microbial fermentation feed efficiently supports production of recombinant protein by Escherichia coli

PONE-D-21-40475R1

Dear Dr. O'Connell

We’re pleased to inform you that your manuscript has been judged scientifically suitable for publication and will be formally accepted for publication once it meets all outstanding technical requirements.

Kind regards,

Vasu D. Appanna

Academic Editor

PLOS ONE

Additional Editor Comments (optional):

Reviewers' comments:

Reviewer's Responses to Questions

**Comments to the Author**

1. If the authors have adequately addressed your comments raised in a previous round of review and you feel that this manuscript is now acceptable for publication, you may indicate that here to bypass the “Comments to the Author” section, enter your conflict of interest statement in the “Confidential to Editor” section, and submit your "Accept" recommendation.

Reviewer #1: All comments have been addressed

Reviewer #2: All comments have been addressed

2. Is the manuscript technically sound, and do the data support the conclusions?

Reviewer #1: Yes

Reviewer #2: Yes

3. Has the statistical analysis been performed appropriately and rigorously? 

Reviewer #1: Yes

Reviewer #2: Yes

4. Have the authors made all data underlying the findings in their manuscript fully available?

Reviewer #1: Yes

Reviewer #2: Yes

5. Is the manuscript presented in an intelligible fashion and written in standard English?

Reviewer #1: Yes

Reviewer #2: Yes

6. Review Comments to the Author

Reviewer #1: The authors have properly addressed my comments, I would like to suggest the manuscript to be accepted at this present version.

Reviewer #2: (No Response)

7. PLOS authors have the option to publish the peer review history of their article (what does this mean?). If published, this will include your full peer review and any attached files.

Reviewer #1: No

Reviewer #2: **Yes: **Noureddine Bouaïcha

---

## [Editor Report · Acceptance letter]

14 Apr 2022

PONE-D-21-40475R1 

Conversion of mammalian cell culture media waste to microbial fermentation feed efficiently supports production of recombinant protein by *Escherichia coli*

Dear Dr. O'Connell:

I'm pleased to inform you that your manuscript has been deemed suitable for publication in PLOS ONE. Congratulations! Your manuscript is now with our production department. 

Kind regards, 

on behalf of

Dr. Vasu D. Appanna 

Academic Editor

PLOS ONE